# VEGF/VEGF-R/RUNX2 Upregulation in Human Periodontal Ligament Stem Cells Seeded on Dual Acid Etched Titanium Disk

**DOI:** 10.3390/ma13030706

**Published:** 2020-02-05

**Authors:** Francesca Diomede, Guya Diletta Marconi, Marcos F. X. B. Cavalcanti, Jacopo Pizzicannella, Sante Donato Pierdomenico, Luigia Fonticoli, Adriano Piattelli, Oriana Trubiani

**Affiliations:** 1Department of Medical, Oral and Biotechnological Sciences, University “G. D’Annunzio”, Chieti-Pescara, 66100 Chieti, Italy; francesca.diomede@unich.it (F.D.); guya.marconi@unich.it (G.D.M.); sante.pierdomenico@unich.it (S.D.P.); luigia.fonticoli@gmail.com (L.F.); adriano.piattelli@unich.it (A.P.); 2Biophotonics Laboratory, Nove de Julho University, São Paulo 01506-000, Brazil; mxistocavalcanti@gmail.com; 3Dental Clinic, Nove de Julho University, São Paulo 01506-000, Brazil; 4ASL 02 Lanciano-Vasto Chieti, «Ss. Annunziata» Hospital, 66100 Chieti, Italy; jacopo.pizzicannella@unich.it

**Keywords:** osseointegration, osteogenesis, angiogenesis, cytocompatibility

## Abstract

In restorative dentistry, the main implants characteristic is the ability to promote the osseointegration process as the result of interaction between angiogenesis and osteogenesis events. On the other hand, implants cytocompatibility remains a necessary feature for the success of surgery. The purpose of the current study was to investigate the interaction between human periodontal stem cells and two different types of titanium surfaces, to verify their cytocompatibility and cell adhesion ability, and to detect osteogenic and angiogenic markers, trough cell viability assay (MTT), Confocal Laser Scanning Microscopy (CLSM), scanning electron microscopy (SEM), and gene expression (RT-PCR). The titanium surfaces, machined (CTRL) and dual acid etched (TEST), tested in culture with human periodontal ligament stem cells (hPDLSCs), were previously treated in two different ways, in order to evaluate the effects of CTRL and TEST and define the best implant surface. Furthermore, the average surface roughness (Ra) of both titanium surfaces, CTRL and TEST, has been assessed through atomic force microscopy (AFM). The vascular endothelial growth factor (VEGF) and Runt-related transcription factor 2 (RUNX2) expressions have been analyzed by RT-PCR, WB analysis, and confocal laser scanning microscopy. Data evidenced that the different morphology and topography of the TEST disk increased cell growth, cell adhesion, improved osteogenic and angiogenic events, as well osseointegration process. For this reason, the TEST surface was more biocompatible than the CTRL disk surface.

## 1. Introduction

Adult stem cells, or somatic stem cells, are immature cells presents in different adult tissues [1]. They are able to differentiate into mature cells with peculiar properties [2]. The main tasks of mature stem cells are to sustain and restore tissues in which the cells are detected, as well as to preserve the stem cell population. Stem cells can be extracted from different tissues such as bone marrow, skeletal muscle, cartilage, dental organ, adipose tissue, synovium, and cardiac tissue [3]. The major source of stem cells that has been widely investigated is the bone marrow, which consists of hematopoietic stem cells and non-hematopoietic cells [4]. Recently, human mesenchymal stem cells (hMSCs) have been playing a pivotal role in regenerative medicine and tissue engineering.

These cells have demonstrated a capacity to repair, replace, and regenerate cells, tissues, and organs [5]. To date, the evidence of the therapeutic capabilities of hMSCs has produced strong optimism: they could be utilized as a ‘multi-talent cell source’ for cell therapeutics with notable clinical applications [6]. Promising early clinical proof-of-concept investigations for treatment of graft-versus-host illness have been reported.

As widely reported in the literature, the human periodontal ligament stem cells (hPDLSCs) are the most used hMSCs population in the restorative dentistry field. They are extracted from the periodontal ligament, a specialized connective tissue that links the alveolar bone socket with the tooth root surface, characterized by different cell populations as osteoblasts, fibroblasts, epithelial cells, endothelial cells, and stem cells [7]. The hPDLSCs have multipotent and proliferative features [8]. Actually, hPDLSCs improve the formation of an interface between dental devices used for implants and bone during the osseointegration process in which a key role is played by the formation of new blood vessels. Vascularization is dependent on vascular endothelial growth factor (VEGF), which promotes both angiogenesis and osteogenesis. In the last two decades, VEGF was the earliest protein displaying a pivotal role pairing osteogenesis and angiogenesis. VEGF’s inactivation concurrently monitored blood vessel invasion and bone formation [9]. For this reason, changes in vascular formation can affect the physiological bone healing, e.g., promoting osteonecrosis, osteoporosis, and non-union fractures [10].

Studies of the last few years confirm that angiogenesis is closely connected with osseointegration, and hMSCs together with endothelial cells have been found to improve the vasculogenic processes [11]. Vascular endothelial growth factor is not solely an important activator of endothelial growth and vascular formation, but also has direct and indirect effects on bone formation by influencing different cell types implicated in the event. The hMSCs, osteoprogenitors, osteoblasts, and osteoclasts all express both VEGF and VEGF receptors and reply to VEGF signaling by promoting recruitment, differentiation, and activity [12].

The noteworthy role of angiogenesis in regenerative dental practices, which deal with dentin–pulp complex and dental pulp regeneration, has been newly evaluated [13,14]. The growth of novel bones, bone regeneration, and also osseointegration after dental implant installation necessitate a blood supply, in order to provide nutrition, oxygen, and osteoprogenitor cells. The authors have earlier reported that there exists an interaction between angiogenesis and osteogenesis: it is well known that the regulation of angiogenesis exhibits a main role in bone remodeling through the wound healing event [10,15]. Based on the literature, Runt-related transcription factor 2 (RUNX2) has been assessed to control VEGF expression in chondrocytes during endochondral bone formation [16]. RUNX2 is implicated in the modulation of generation of chondrocytes and osteoclasts, activates angiogenesis, enhances the osteogenic microenvironment, and promotes osteogenesis and bone growth [17]. RUNX2 is one of the major osteogenic transcriptional factors, exhibits an essential role in the initial osteogenic differentiation, and it also represents a premature marker of osteogenic differentiation [18].

In the field of implant dentistry, titanium and its alloys are one of the major used materials, due to their distinctive biocompatibility, mechanical features, and chemical stability [19]. The biocompatibility of titanium is associated to its surface characteristics such as its chemical composition, surface roughness, and surface energy. Several attempts have been made to ameliorate the surface characteristics of titanium implants and as a consequence to improve initial bone bonding [20]. Thanks to these features, titanium seems to regulate protein absorption to the metal surface and to improve cell growth and their differentiation [21]. These distinctive characteristics of titanium are connected with the formation of an oxide layer on the surface, which plays a key role in osseointegration [22]. In addition to the TiO_2_ layer, other elements are important for osseointegration, including the titanium implant surface composition and topography [23]. For this reason, more advantageous studies to change titanium surface characteristic are needed to promote bone formation.

Due to the existing interaction between angiogenesis, osteogenesis, and osseointegration, in the present study, two titanium surfaces, machined (CTRL) and dual acid etched (TEST), have been studied [24].

The goal of the present work was to analyze if and how the changes in the surface characteristics of the titanium could affect/improve angiogenic and osteogenic processes, cytocompatibility, angiogenic and osteogenic markers, such as RUNX2 and VEGF, and their receptors.

The null hypothesis of the current study was to obtain no variations in RUNX2 and VEGF expression in hPDLSCs cultured with CTRL and TEST titanium surfaces.

## 2. Materials and Methods 

### 2.1. Ethic Statement

All subjects gave their informed consent for inclusion before they participated in the study. The study was conducted in accordance with the Declaration of Helsinki, and the protocol was accepted by the Ethics Committee of by Medical Ethics Committee at the Medical School, “G. d’Annunzio” University, Chieti, Italy (n°266/17.04.14).

The Department of Medical, Oral and Biotechnological Sciences and the Laboratory of Stem Cells and Regenerative Medicine are certified according to the quality standard ISO 9001:2015 (certificate n°32031/15/S).

### 2.2. Cell Culture

Five human periodontal ligament biopsies were scraped from human premolar teeth of patients generally in good health. The tissue was procured by scaling the roots utilizing Gracey’s curettes [25]. The samples were washed five times with PBS (Lonza, Basel, Switzerland), and cultured utilizing TheraPEAK™MSCGM-CD™ BulletKit serum free, chemically defined (MSCGM-CD) medium for the growth of human Mesenchymal Stem Cells (Lonza, Basel, Switzerland) [26]. The medium was replaced twice a week, and cells migrating from the explants tissue after reaching about 80% of confluence were trypsinized (Lonza, Basel, Switzerland), and then were subcultured until passage 2nd (P2).

### 2.3. Dental Implants

In the present study, two different titanium disks surfaces (Resista, Omegna, VB, Italy) have been used: machined (CTRL) and dual acid etched (TEST). The disks were made of grade 4 titanium.

Two hundred fifty-two samples made of grade 4 titanium (Resista, Omegna [VB]) were machined to procure 8 × 3 mm cylinders. Rough surfaces were made starting from titanium by dual acid etching (TEST) [27].

### 2.4. Atomic Force Microscopy Observations (AFM)

The morphologies of two disks surfaces, machined (CTRL) and dual acid etched (TEST), were analyzed by Atomic Force Microscopy (AFM), utilizing a Multimode 8 Bruker AFM microscope (Bruker, Milan, Italy) coupled with a Nanoscope V controller (Bruker AXS, Marne La Vallee, France) and commercial silicon tips (RTESPA 300, resonance frequency of 300 kHz and nominal elastic constant of 40 N/m) were utilized in ScanAsyst air mode. By the means of this mode, it was acceptable, from the height panel, to measure roughness. Then the software Nanoscope was used to analyze images and 3D reconstruction. The roughness average (Ra), that is the arithmetic mean of the absolute values of the height of the surface profile, was considered for the statistical analysis. Four samples of each group were observed and the mean values (+/− standard deviation) were considered for the statistical analysis [28].

### 2.5. Scanning Electron Microscopy (SEM) Analysis

CTRL and TEST samples were seeded with hPDLSCs for 21 days and then were fixed for 4 h at 4 °C in 4% Glutaraldehyde in 0.05 M phosphate buffer (pH 7.4), dehydrated in increasing ethanol concentrations and afterward critical point dried. They were then mounted on aluminum stubs and gold-coated in an Emitech K550 (Emitech Ltd., Ashford, UK) sputter-coater before imaging by means of a SEM (ZEISS, EVO 50, Jena, Germany) [29].

### 2.6. Trough Cell Viability Assay (MTT) Assay

The cell viability of hPDLSCs cultured with or without CTRL and TEST samples was assessed by the quantitative colorimetric MTT (3-[4,5-dimethyl-2-thiazolyl]-2,5-diphenyl-2H-tetrazoliumbromide test) (Promega, Milan, Italy) as previously reported [30]. A total of 2.5 × 10^5^ cells/well were seeded into a 96-well culture plate with MSCBM medium (Lonza), after 24 h of incubation at 37 °C, 15 μl/well of MTT was added to culture medium, and cells were incubated for 3 h at 37 °C. The supernatants were read at 650 nm wavelength utilizing an ND-1000 NanoDrop Spectrophotometer (NanoDrop Technologies, Rockland, DE, USA). The MTT test was assessed in three independent analysis.

### 2.7. Confocal Laser Scanning Microscopy (CLSM) Analysis

Cells cultured on CTRL and TEST tested surfaces were fixed for 10 min at room temperature (RT) with 4% paraformaldehyde in 0.1 M sodium phosphate buffer (PBS), pH 7.4 [31]. Successively washing in PBS, cultures were prepared for immunofluorescence labeling. The hPDLSCs cultured on the titanium surfaces were permeabilized with 0.1% Triton X-100 in PBS, followed by blocking with 5% skimmed milk in PBS. Primary monoclonal antibodies to anti human VEGF (Santa Cruz Biotechnology, Santa Cruz, CA; USA), VEGF-R (Santa Cruz Biotechnology), and RUNX2 (Santa Cruz Biotechnology) were utilized, followed by Alexa Fluor 488 green fluorescence conjugated goat anti-mouse as secondary antibodies (Molecular Probes, Invitrogen, Eugene, OR, USA). Successively, samples were incubated with Alexa Fluor 594 phalloidin red fluorescence conjugate (Molecular Probe), as an actin cytoskeleton marker. Nuclei were dyed with TOPRO (Molecular Probe). Samples were put down on glass slides and mounted with Prolong antifade (Molecular Probes) [32]. Dyeing of samples was seen utilizing a Zeiss (Jena, Germany) LSM510 META confocal system, connected to an inverted Zeiss Axiovert 200 microscope supplied with a Plan Neofluar oil-immersion objective (40×/1.3 NA). The pictures were taken utilizing an argon laser beam with excitation lines at 488 nm.

### 2.8. Gene Expression

VEGF and RUNX2 mRNA expression were executed by real-time PCR. Successively, total RNA was isolated utilizing the Total RNA Purification Kit(NorgenBiotek Corp., Ontario, CA) as stated by the manufacturer’s instructions [33]. The M-MLV Reverse Transcriptase reagents (Applied Biosystems) were utilized to create cDNA. Real-Time PCR was executed with the Mastercycler ep realplex real-time PCR system (Eppendorf, Hamburg, Germany). The Expression levels in cells cultured on CTRL and TEST samples were assessed. Commercially accessible TaqMan Gene Expression Assays (VEGF Hs00900055_m1; VEGF-R Hs00157093_m1 and RUNX2 Hs00231692_m1) and the TaqMan Universal PCR Master Mix (Applied Biosystems, Foster City, CA, USA) were utilized according to standard protocols [34]. Beta-2 microglobulin (B2M Hs99999907_m1) (Applied Biosystems, Foster City, CA, USA) was utilized for template normalization. RT-PCR was analyzed in three independent analysis; duplicate determinations were obtained for each sample.

### 2.9. Protein Expression

Thirty micrograms of proteins obtained from all samples were processed as previously reported [35].

Membranes were incubated with primary antibodies rabbit anti-VEGF (1:750, rabbit; Sigma-Aldrich, Milan, Italy), RUNX2 (1:750, rabbit; Sigma-Aldrich), and anti-beta-actin (1:750, mouse; Santa Cruz Biotechnology, Santa Cruz, CA). After five washes in PBS 0.1% Tween-20, the tested samples were incubated for 1 h at room temperature with peroxydase-conjugated secondary antibody anti-rabbit and anti-mouse diluted 1:2000 in 1X PBS, 3% milk, 0.1% Tween [36]. Protein expression was analyzed using the enhanced chemiluminescence detection system (ECL) (Amersham Pharmacia Biotech, PA, USA) with photo documenter Alliance 2.7 (Uvitec, Cambridge, UK). Signals were taken by ECL enhancing and assessed utilizing an UVIband-1D gel analysis (Uvitec).

### 2.10. Data and Statistical Analysis

The Statistical Package for Social Science (SPSS, v.21.0, Inc., Chicago, IL, USA) was utilized for data analysis. Parametrical methods were utilized after having verified the existence of the required assumptions. In detail, the normality of the distribution and the equality of variances were analyzed by the Shapiro–Wilk and Levene’s tests, respectively. Data were expressed as means and standard deviation of the recorded values. The differences among the levels of the factors under investigation were obtained performing three distinct two-way-ANOVA tests, one for each experiment. Tukey tests were performed for pairwise comparisons. A value of *p* < 0.05 was considered statistically significant in all tests.

## 3. Results

### 3.1. Atomic Force Microscopy Measurements

The two titanium surfaces, CTRL and TEST, were assessed by utilizing AFM. In Figure 1, topographical and tridimensional micrographs of the titanium surfaces are analyzed. From the pictures reported, the variations of the topography of the surface CTRL respect to TEST are noticeable. From the height panel, the Nanoscope analysis 1.8 software (Bruker, Milan, Italy) is able to evaluate the roughness [37]. The AFM observations have shown that the TEST group was characterized by an average roughness (Ra) of 109.6 ± 35.7 nm and 367 ± 116.9 nm (Ra) for the CTRL (Figure 1). These data obtained evidenced that the roughness for the TEST surface is strongly higher with respect to the CTRL surface.

### 3.2. Human PDLSCs Adhesion

SEM observations showed the surface morphology of two different considered surfaces, CTRL and TEST, without cells (Figure 2A,B). Human PDLSCs seeded on TEST for 24 h showed a better ability of adhesion compared to CTRL surface (Figure 2C,D).

### 3.3. Proliferation Assay of hPDLSCs Cultured on CTRL and TEST Titanium Surfaces

CTRL and TEST surfaces do not modify the cell proliferation rate, as evidenced in the graph. hPDLSCs seeded on both surfaces reported a similar proliferation rate. Overlapping results were obtained by means of MTT assay and Trypan Blue exclusion test (Figure 3).

### 3.4. Titanium Surfaces Influence Protein Expression at CLSM

Figure 4, Figure 5, Figure 6 and Figure 7 showed fluorescence images of the cytoskeleton actin (phalloidin, green) and the nuclei (TOPRO, blue) of hPDLSCs cultured on CTRL and TEST group of samples taken after 1 and 8 weeks of culture. The cells adhered and spread well with a spindle fibroblast-like shape on all samples, which indicates that the different surface treatment did not affect the cytocompatibility. The number of cells increased from 1 week to 8 weeks (Figure 4, Figure 5, Figure 6 and Figure 7). The CLSM observation demonstrated the tendency of cells seeded on CTRL and TEST surfaces to differentiate versus the osteogenic lineage after 1 and 8 weeks of culture, which can be confirmed by RUNX2 positive expression (Figure 5, Figure 6 and Figure 7). The hPDLSCs cultured on TEST implant surface evidenced an higher expression of VEGF and VEGFR after 8 weeks of culture compared to cells seeded on CTRL after 8 weeks. Furthermore, hPDLSCs seeded on both implant surfaces after 1 week of culture did not evidence any VEGF expression, but exclusively VEGFR expression is reported (Figure 4, Figure 5, Figure 6 and Figure 7).

### 3.5. VEGF and RUNX2 Expression

Bar graphs showed the gene expression of VEGF and RUNX2 assessed by RT-PCR after 8 weeks of culture (Figure 7A). Human PDLSCs cultured on TEST evidenced an higher expression of VEGF and RUNX2 compared to hPDLSCs cultured on CTRL surface. Gene expression confirmed the qualitative results obtained by CLSM observations. Protein expression of specific bands of VEGF, a major contributor of angiogenesis, and RUNX2 showed an over expression in hPDLSCs cultured on TEST compared to the cells seeded on CTRL surface (Figure 7B). Moreover, the densitometric analysis, the related protein bands quantification normalized with β-Actin, reported a similar trend obtained by gene expression results (Figure 8C). These data confirmed the obtained gene expression results.

## 4. Discussion

In the last few years, several researches have been focused on detection of biomaterials features able to reduce the percentage of dental implants failures. Different treatment techniques on different materials have been tested to ensure a good interaction between bone and implants surfaces and to promote faster healing. The main goal of recent researches is to identify a material with a high performance, able to guarantee direct contact between bone and implant, and to absorb biological molecules on its surface [38]. Concerning titanium, a great number of methods have been introduced to modify implants surface, such as physicochemical, morphological, and biochemical techniques [39]. Acid etching and sandblasting have been reported to produce high-performing surfaces [40].

Chemical alterations of shape, edge, roughness, and wettability of the surface could influence the ability to absorb protein and consequently to improve the healing processes at the interface [41,42]. Based on the literature, an important role is played by mesenchymal stem cells (MSCs) in these processes [43,44]. The decision of choosing hMSCs is led by the necessity to identify a favorable experimental model to assess the osseointegration and healing events happening during the routine implantology and reconstructive or regenerative surgical methods. For this reason, the use of hMSCs seeded with the titanium implants constitutes an appropriate and worthy experimental method to better comprehend the mechanisms of cell interaction with diverse implant surfaces [45].

Taking into consideration titanium features and hPDLSCs proprieties, it was decided to investigate the osseointegration and angiogenesis involved in bone formation using an in vitro model of hPDLSCs seeded on two different titanium surfaces.

In the current work, the roughness of two titanium surfaces, CTRL and TEST, was analyzed by AFM. The AFM observations have reported that the roughness for the TEST surface is notably higher compare to the CTRL surface. Several works supported that moderately rough surfaces are capable of inducing more rapid osseointegration with respect to turned implant surfaces [46].

Successively, the hPDLSCs have been cultured on two different surfaces, CTRL and TEST, respectively. Even if, based on the literature, several methods have been standardized to assess cell viability, in the present experimental model the MTT assay has been performed [47]. The cell viability assay evidenced a similar trend in cells cultured with CTRL and TEST surface. Moreover, under SEM observation, no significant differences have been identified between hPDLSC/CTRL and hPDLSCs/TEST surface. The hPDLSCs showed the same morphology and adhesion ability on both surfaces. Data obtained from western blot analysis and CLSM observation, after 8 weeks of culture, showed an upregulation of RUNX2, VEGF, and VEGFR in cells cultured on both titanium surfaces compared to cells not cultured on disks. In detail, RUNX2 and VEGF were significantly expressed in hPDLSCs cultured on TEST respect to CTRL. On the other hand, after 1 week of culture, VEGF expression has not been observed under CLSM analysis. The present data, obtained under CLSM, evidenced an overexpression of RUNX2, an early osteogenic marker, and VEGF, an angiogenic protein, in TEST compared to CTRL sample. These results lead us to hypothesize that the TEST surface had better osteogenic and angiogenic properties. To further confirm these results, RT-PCR was performed. The hPDLSCs seeded on TEST surfaces, after 8 weeks of culture, exhibited an upregulation of VEGF and RUNX2 compared to hPDLSCs seeded on CTRL surfaces. These evidences confirmed the data obtained by Western Blot and under CLSM observations. Interestingly, earlier studies reported that RUNX2 was a part of the genetic program that regulated the expression of VEGF during endochondral bone formation [48]. The present data could suggest that the increase of RUNX2 could promote VEGF and VEGF-R expression. As widely reported in the literature, VEGF regulates endochondral bone formation, in fact downregulation of VEGF expression results in negative consequences on angiogenesis, indirectly on osteogenesis, and on the whole healing process [12]. Angiogenesis is directly related to the osteogenesis process, due to the importance of blood vessels in the formation and maintenance of new bone [49]. Furthermore, VEGF has also been demonstrated to regulate calvaria ossification [50]. Many studies reported that the reduction of the ossification process in calvaria was related to the loss of VEGF expression in osteoblast progenitor cells [51]. For this reason, rapid formation of new vessels in the area of surgery is of fundamental importance in promoting bone regeneration. Both disk surfaces, in contact with hPDLSCs, have revealed not only osteoconductive characteristics, assessed through the cell adhesion and proliferation, but also the capacity to induce VEGF release from hPDLSCs, as shown by gene expression and under confocal microscopy. The induction of the production of VEGF and VEGF receptor from hPDLSCs could symbolize a goal for tissue engineering and, especially, for the therapeutic formation of new blood vessels around the biomaterial in the initial part of osseointegration. Taking into consideration the results obtained, the future prospects could be to evaluate osteogenic and angiogenic events induced by human dental pulp stem cells (hDPSCs) and human gingival mesenchymal stem cells (hGMSCs) on the TEST titanium implant surface.

## 5. Conclusions

Despite the limitations of the present in vitro study, the following conclusions could be outlined. To conclude, the present results have shown an important role of VEGF and its receptor, and a better performance of TEST surfaces when compared to CTRL surfaces. The capability to enhance the levels of VEGF and its receptor could lead to more rapid bone–titaniumintegration.

## Figures and Tables

**Figure 1 materials-13-00706-f001:**
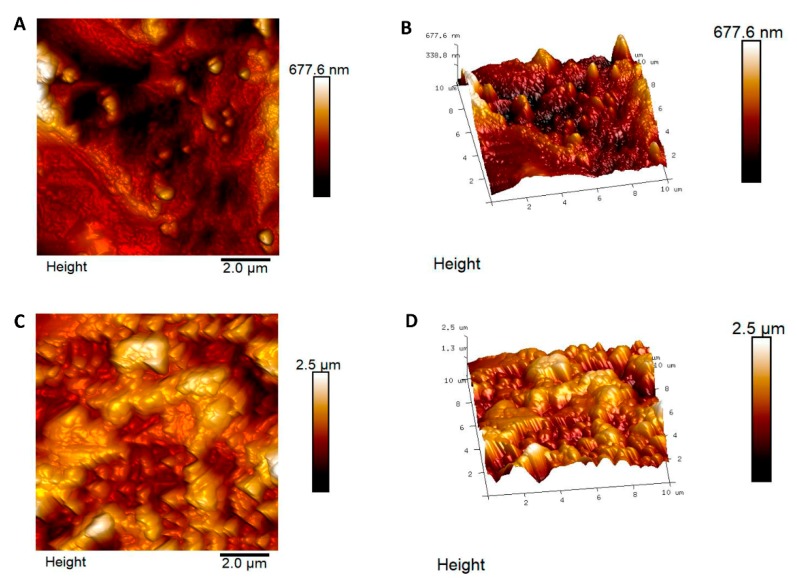
(A,C) Topographical and (B,D) Three-dimensional Atomic Force Miscroscopy (AFM) pictures of machined titanium surface (CTRL, upper line) and dual acid etched titanium surface (TEST, bottom line).

**Figure 2 materials-13-00706-f002:**
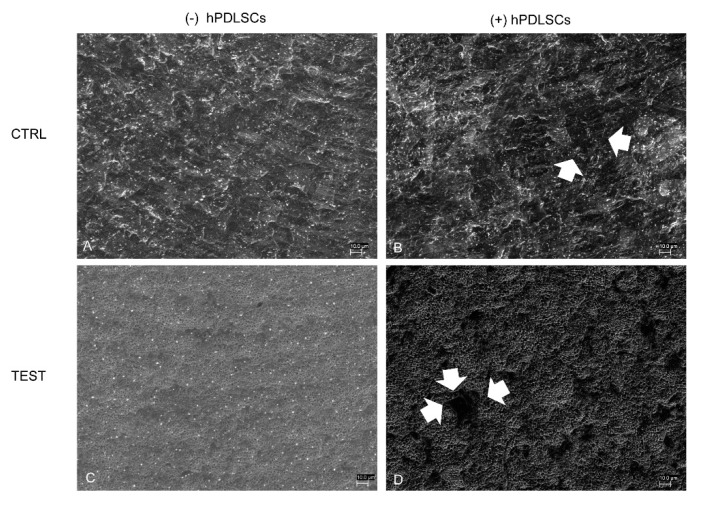
Scanning electron microscopy (SEM) analysis of the human periodontal ligament stem cells (hPDLSCs) cultured on the titanium implant surfaces. (**A**) CTRL surface without cells. (**B**) TEST surface without cells. (**C**) Human PDLSCs adhere on CTRL surface. (**D**) Human PDLSCs adhere on TEST surface. Mag: 1000×.

**Figure 3 materials-13-00706-f003:**
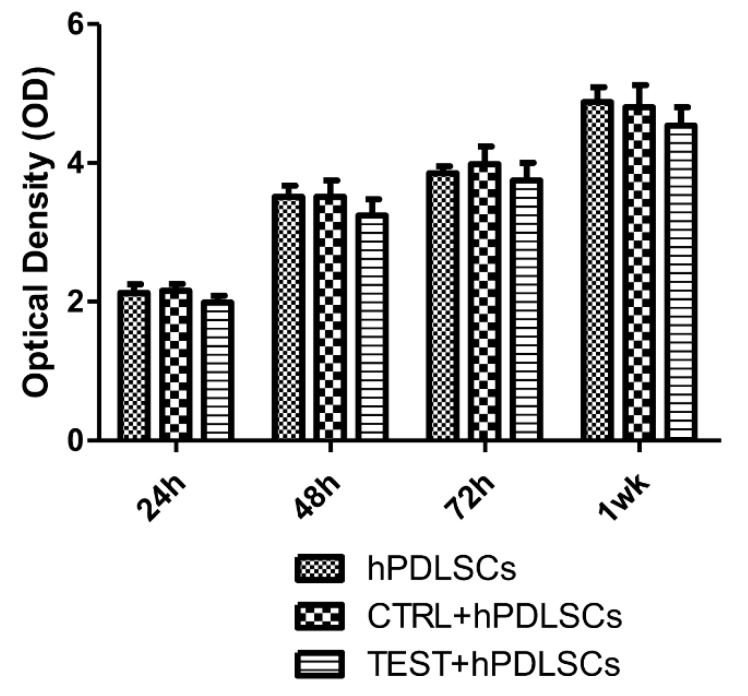
Cell viability assay of hPDLSCs cultured on CTRL and TEST titanium surfaces. Trough cell viability assay (MTT) assay was performed at 24, 48, 72 h, and 1 week on both CTRL and TEST titanium surfaces.

**Figure 4 materials-13-00706-f004:**
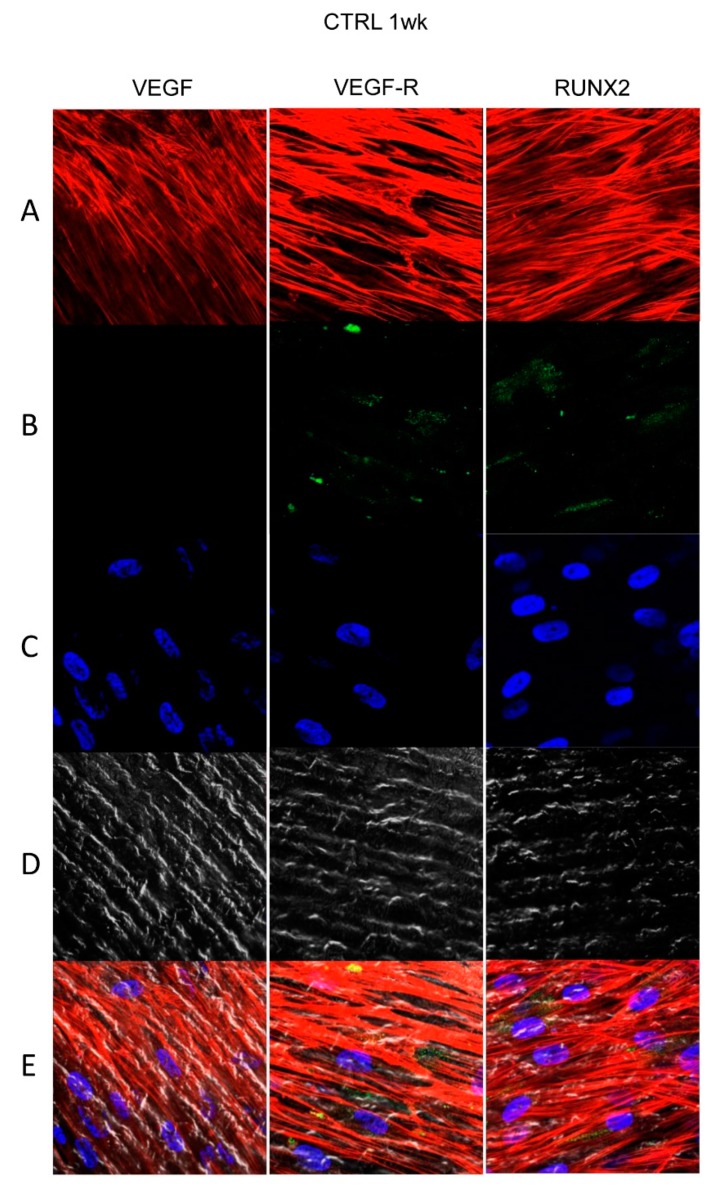
Human PDLSCs cultured on CTRL titanium implant surface were observed after 1 week of being cultured. (**A**) Cytoskeleton actin was dyed in red fluorescence; (**B**) specific markers (vascular endothelial growth factor [VEGF], VEGF-R, and Runt-related transcription factor 2 [RUNX2]) were stained in green fluorescence; (**C**) nuclei were dyed in blue fluorescence. (**D**) PDLSCs with CTRL; (**E**) TL, transmission light: gray. Scale bar: 10 µm.

**Figure 5 materials-13-00706-f005:**
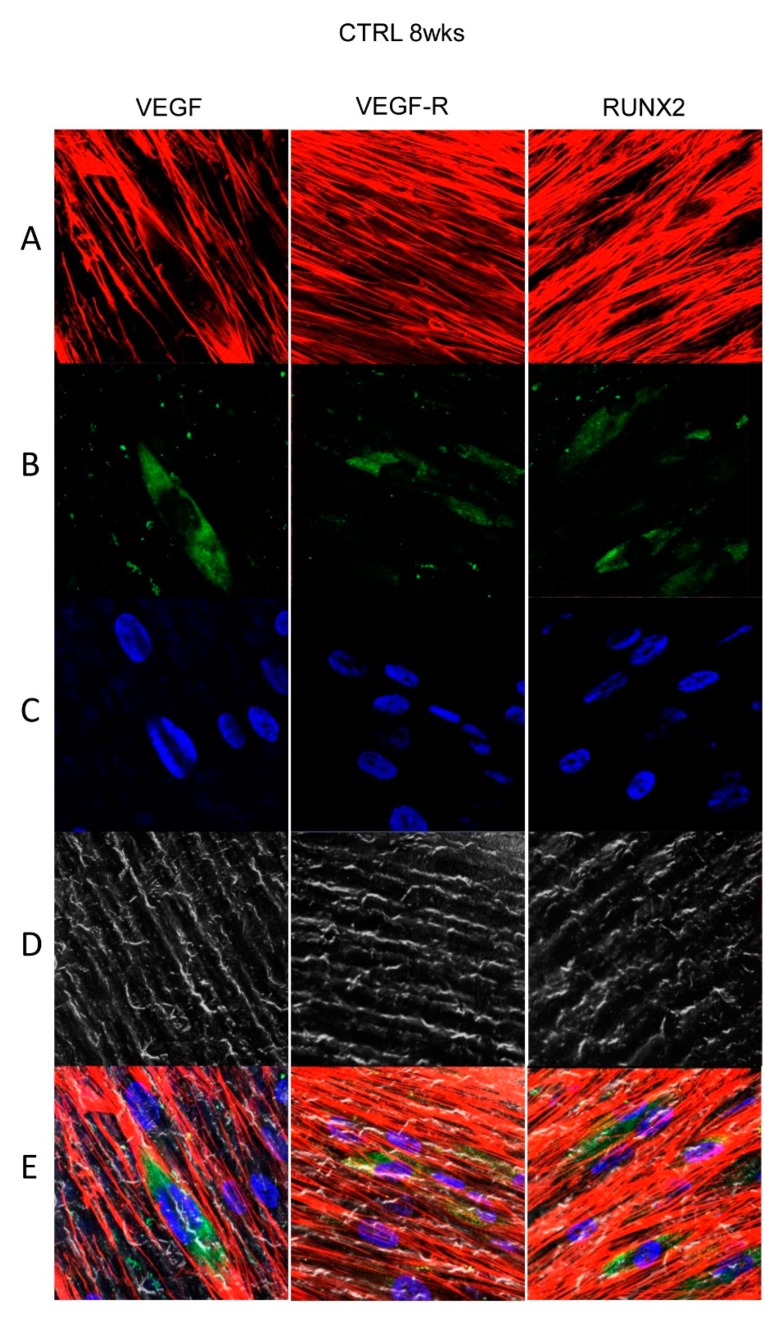
Human PDLSCs cultured on CTRL titanium implant surface were observed after 8 weeks of being cultured. (**A**) Cytoskeleton actin was dyed in red fluorescence; (**B**) specific markers (VEGF, VEGF-R, and RUNX2) were stained in green fluorescence; (**C**) nuclei were dyed in blue fluorescence. (**D**) PDLSCs with CTRL; (**E**) TL, transmission light: gray. Scale bar: 10 µm.

**Figure 6 materials-13-00706-f006:**
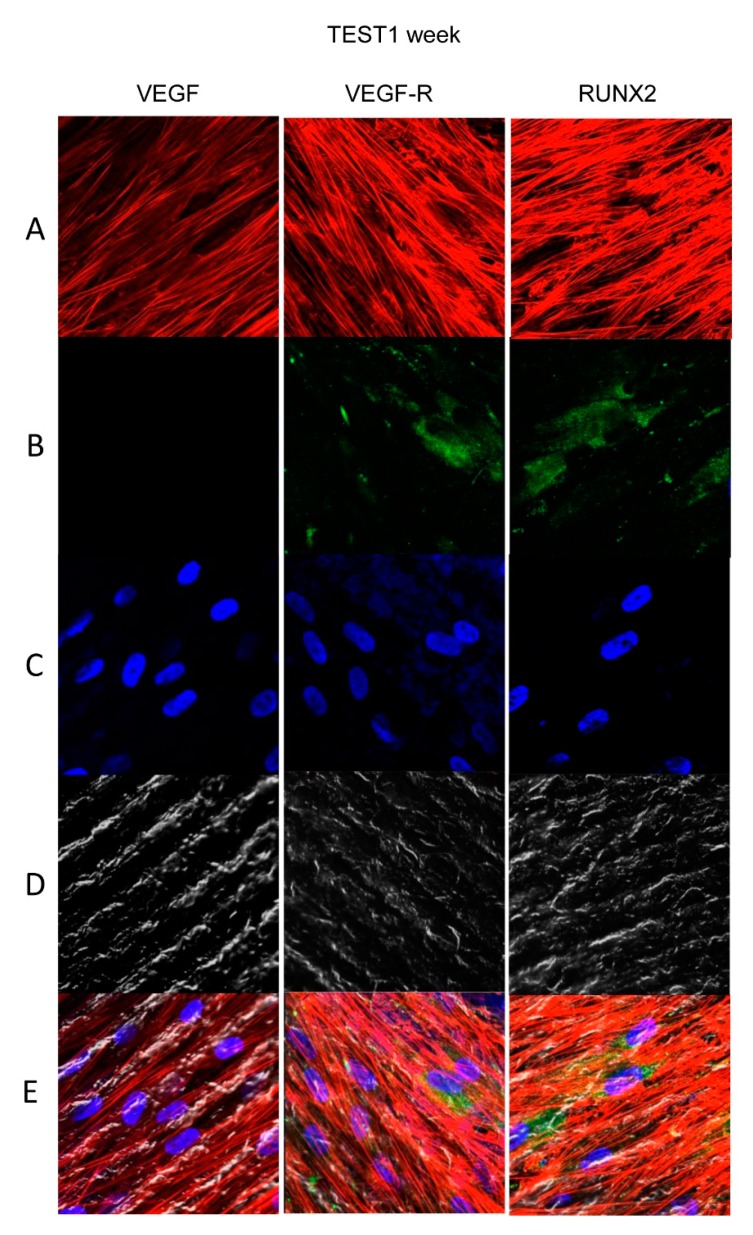
Human PDLSCs cultured on TEST titanium implant surface were observed after 1 week of incubation. (**A**) Cytoskeleton actin was stained in red fluorescence; (**B**) specific markers (VEGF, VEGF-R, and RUNX2) were dyed in green fluorescence; (**C**) nuclei were dyed in blue fluorescence. (**D**) PDLSCs with TEST; (**E**) TL, transmission light: gray. Scale bar: 10 µm.

**Figure 7 materials-13-00706-f007:**
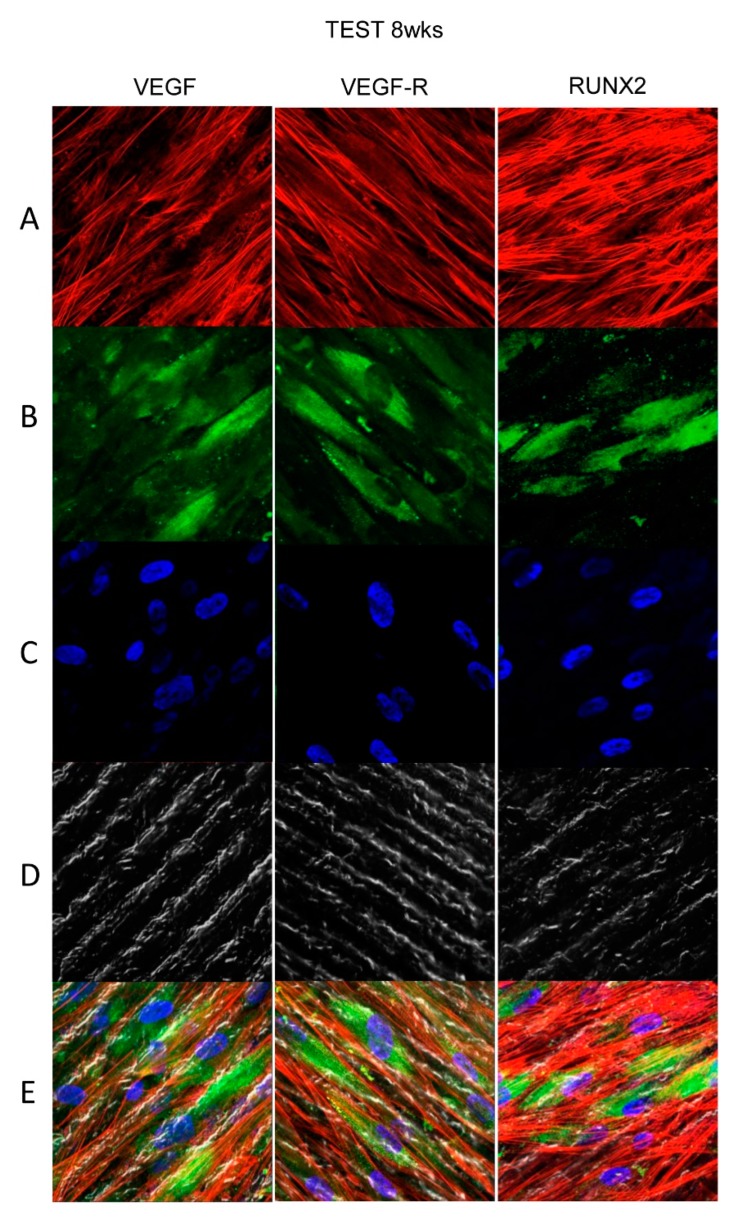
Human PDLSCs cultured on TEST titanium implant surface were observed after 8 weeks of incubation. (**A**) Cytoskeleton actin was dyed in red fluorescence; (**B**) specific markers (VEGF, VEGF-R, and RUNX2) were stained in green fluorescence; (**C**) nuclei were dyed in blue fluorescence. (**D**) PDLSCs with TEST; (**E**) TL, transmission light: gray. Scale bar: 10 µm.

**Figure 8 materials-13-00706-f008:**
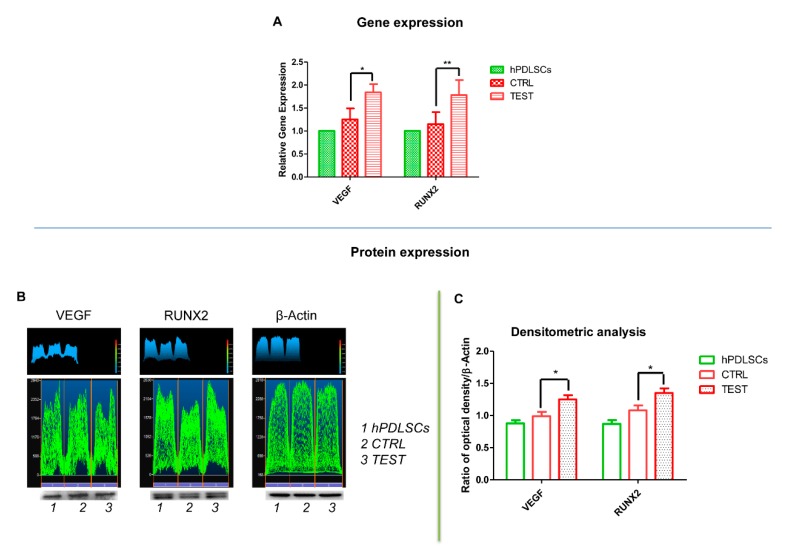
(**A**) Graph bar of RT-PCR of VEGF and RUNX2 in cells cultured on CTRL and TEST surface ** *p* < 0.01. (**B**) Protein-level expression of VEGF and RUNX2 in cells cultured on CTRL and TEST surface. (**C**) Graphs report densitometric measurements of proteins bands expressed as integrated optical intensity (IOI) mean of three separate analysis.

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
