# Peer review of "VEGF/VEGF-R/RUNX2 Upregulation in Human Periodontal Ligament Stem Cells Seeded on Dual Acid Etched Titanium Disk"

_materials, 2020, doi:10.3390/ma13030706_

Round 1

Reviewer 1 Report

1, In the introduction and discussion sections of this article, the authors described the importance of surface characteristics of titanium implant in angiogenesis, osteogenesis, and osseointergration, such as the roughness, chemical properties, surface charge, wettability, etc. However, I think the experimental data on the measurement of the surface properties of materials is only SEM, which I think is not enough. Please add more experiments related to surface characteristics, such as XPS, contact angles, roughness, etc.

2, Please describe the effects of RUNX2 and VEGF on osteogenesis and angiogenesis in the introduction section.

Reviewer 2 Report

This is an interesting scientific work on the regulation of mediators in human periodontal ligament stem cells in relation to titanium surfaces for implant use.
The topic is interesting and in my opinion falls within the purpose of the magazine.
However, there are some critical issues:
-Line 19 You don't start a sentence with "In dentistry"
-In the abstract section a sentence on the techniques used (for example MTT) should be added
-The introduction section appears confusing and not well structured. In particular, the initial part should focus on implant success, the real clinical objective of the materials, and not on stem cells.
-Lines 47-81 the period is too long and without interruptions
-Line 81 Authors should enter null study hypotheses at the end of the Introduction section
-Line 92 Why 5 biopsies?
-Line 102 The reader of an article should have information on the techniques performed and not a simple bibliographical reference. A very concise sentence must be inserted
-Line 108 Missing producer country of the SEM
-Figures: A bad habit of the authors is to comment on the results in the legends of the tables and graphs. These must be absolutely removed. We need a simple and clear legend only of what the reader sees.
-Line 186 Add Figures 3-7 instead of 4-7
-Figure 7 Legend: this legend is absolutely redundant; eliminate the "error barr ... .." ANOVA ", etc ....
-Line 245 Indicate the meaning of MSCs
-All discussion section is too small.
What the recent literature reports on these tests is not taken into consideration by the authors trying to give a critical analysis of the results obtained. In particular, I expect them to be compared, for example the various cytotoxicity techniques in order to define the most suitable one for the specific study. In this regard, I recommend inserting the following scientific work into the references and discussion:
-Pagano S, Lombardo G, Balloni S, Bodo M, Cianetti S, Barbati A, Montaseri A,
Marinucci L. Cytotoxicity of universal dental adhesive systems: Assessment in
vitro assays on human gingival fibroblasts. Toxicol In Vitro. 2019 Jun
11; 60: 252-260. doi: 10.1016 / j.tiv.2019.06.009.
-Line 279 The limitations of the study and the future prospects that must be indicated by the authors are missing.
-References: a very bad habit on the part of the authors is represented by the excessive number of self-citations, not always useful for the discussion of the introductions and discussions sections.
7 works attributable to the authors are too many. Remove the least useful one at work. This represents, for me, a very important criterion for accepting scientific work.

An important english grammar check is requested by a native english teacher.

Round 2

Reviewer 2 Report

All comments were modified ad requested.